# Can the fusion of motion capture and 3D medical imaging reduce the extrinsic variability due to marker misplacements?

Xavier Gasparutto[1]*, Jennifer Wegrzyk[1], Kevin Rose-Dulcina[1], Didier Hannouche[2], Stéphane Armand[1]

**1** Laboratory of Kinesiology Willy Taillard, Geneva University Hospitals and University of Geneva, Geneva, Switzerland, **2** Division of Orthopaedics and Trauma Surgery, Geneva University Hospitals, Geneva, Switzerland

* xavier.gasparutto@hcuge.ch

**Data Availability Statement:** Data are available on Yareta (DOI: 10.26037/yareta: md6wej4guvbufkqge322mqyu5i). The code used

## Abstract

In clinical gait analysis, measurement errors impede the reliability and repeatability of the measurements. This extrinsic variability can potentially mislead the clinical interpretation of the analysis and should thus be minimised. Skin marker misplacement has been identified as the largest source of extrinsic variability between measurements. The goal of this study was to test whether the fusion of motion capture and 3D medical imaging could reduce extrinsic variability due to skin marker misplacement. The fusion method consists in using anatomical landmarks identified with 3D medical imaging to correct marker misplacements. To assess the reduction of variability accountable to the fusion method, skin marker misplacements were voluntarily introduced in the measurement of the pelvis and hip kinematics during gait for two patients scheduled for unilateral hip arthroplasty and two patients that underwent unilateral hip arthroplasty. The root mean square deviation was reduced by -78 ± 15% and the range of variability by -80 ± 16% for the pelvis and hip kinematics in average. These results showed that the fusion method could significantly reduce the extrinsic variability due to skin marker misplacement and thus increase the reliability and repeatability of motion capture measurements. However, the identification of anatomical landmarks via medical imaging is a new source of extrinsic variability that should be assessed before considering the fusion method for clinical applications.

## Introduction

In clinical settings, human movement analysis has become an essential tool to identify pathology of the locomotor system, to assess the efficacy of a rehabilitation protocol or a surgery or to follow the evolution of a disease [1]. In all these cases, a key factor to understand properly the patient's pathology is to assess the differences between multiple measurements, e.g. the differences before and after treatment. To understand properly those differences, the variability of the measurement of human motion needs to be taken into account. This variability can be divided into two categories [2,3]: the intrinsic variability due to the inherent variation of the

for the data on the paper are available on GitLab, the relevant links are in the body of the paper.

**Funding:** This project was funded by the "Fondation pour la recherche ostéo-articulaire" of Geneva. The funders had no role in study design, data collection and analysis, decision to publish, or preparation of the manuscript.

**Competing interests:** The authors have declared that no competing interests exist.

human movement and the extrinsic variability due to measurement errors. The extrinsic variability impedes the reliability and repeatability of the measurements and can potentially mislead the clinical interpretation of the analysis. Consequently, extrinsic variability should be minimised.

For measurement based on skin marker methods, the extrinsic variability has three main origins: instrumental error, soft tissue artefacts and skin marker misplacement [4–6]. The latter has been identified as the largest source of extrinsic variability between measurements [7,8]. Multiple methods have been proposed to correct this source of errors such as: palpation methods [9], real-time feedback methods [10,11], external devices [12,13], fusion with static 3D medical imaging [14,15] and dynamic 3D imaging such as bi-plane fluoroscopy [16,17].

Dynamic 3D imaging was used to measure kinematics of joints [16,17] and drive musculoskeletal models with success [18]. Despite their interest for research purpose, those methods are highly invasive due to the large dose of radiations received by the patients and are not applicable in daily routine. Static medical imaging methods have been used to personalize musculoskeletal models [19] and, the fusion of motion capture with low-dose static bi-plane X-rays has been advocated to estimate the hip joint centre in clinical gait analysis [15]. The fusion method consists in identifying the relative position of the skin markers (external markers) and the underlying anatomical landmarks (internal markers) with 3D medical imaging. Then, the trajectories of the internal markers are estimated from the trajectories of the external markers measured during the tasks by the motion capture system. The reconstruction of the internal markers is based on the hypothesis that the transformation from the external markers to the internal markers is constant. This method could potentially remove the extrinsic variability due to external marker misplacement but it will not address the issue of soft tissue artefacts. To our knowledge, bi-plane X-rays has only been used as a gold standard to validate marker-based methods that identify anatomical landmarks [14,20,21] but its potential to reduce the variability due to external marker misplacements has not been assessed yet.

The aim of this study was to test whether the fusion of motion capture and 3D medical imaging could reduce the extrinsic variability due to external marker misplacement on the pelvis in clinical gait analysis. To that end, a controlled extrinsic variability was introduced in the measurement of gait by adding external markers intentionally misplaced on the pelvis to a markerset positioned through palpation. We hypothesized that the pelvis and hip kinematic variability computed by the fusion method will be significantly lower than the variability computed directly from the external markers.

## Material & methods

The "Commission cantonale d'éthique de la recherche" of Geneva approved this study (CCER-2017-00817) and all participants gave written informed consent. The workflow from the measurements to the outcome of the study is presented in Fig 1.

### Participants

Four patients participated in this study (69.7±9.4 years old, 60.5±11.7 kg, 1.65±0.02 m). Two out of the four patients were scheduled for a total hip arthroplasty (patients (b) and (d)) and the other two underwent the surgical procedure 3 months before the measurement (patients (a) and (c)). The measurements were performed by one operator and the data used for this paper can be found with the following DOI: 10.26037/yareta:md6wej4guvbufkqge322mqyu5i.

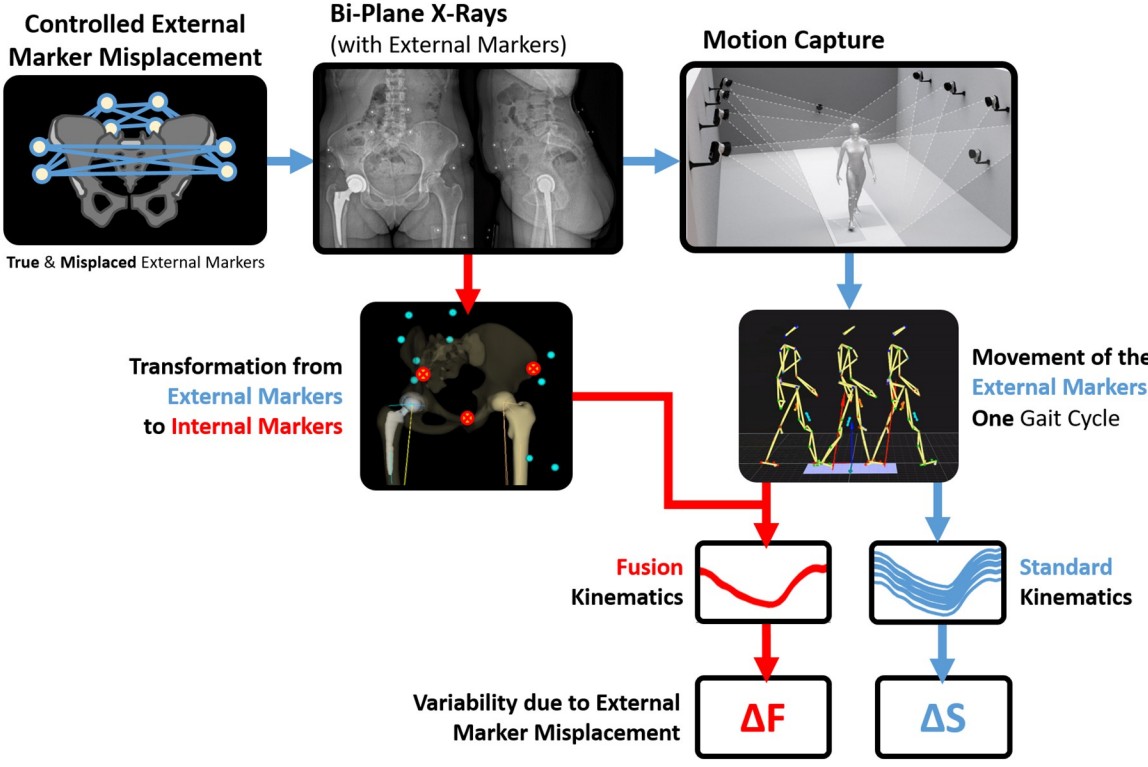

**Fig 1. Workflow of the standard method and fusion method.**

## Controlled marker misplacement error

The patients were equipped with four external markers placed by palpation [9] on the right and left anterior superior iliac spines (RASI, LASI), and on the right and left posterior superior iliac spines (RPSI, LPSI). Four additional external markers were intentionally misplaced on the pelvis to introduce a controlled skin marker misplacement error in the measurement. Each patient had one configuration of marker misplacement. The four marker misplacement configurations (Fig 2) were chosen to impact different angles of the pelvis: a) vertical offset, b) axial rotation, c) anteversion, and d) obliquity. The palpated markers will be referred to as the "true" external markers in opposition to the "misplaced" external markers. Each misplaced marker was associated with a true marker and was placed at a distance of two skin markers from the true marker. The external markers positions were marked with a surgical marking pen (Medeco-ch, Duillier, Switzerland) before taping them to the skin, in case a marker would fall. Lead beads (Split Shot, 0.29 g, Caperlan, Cestas, France) were fixed at the centre of the markers to facilitate their identification on the bi-plane X-ray images.

## Bi-plane X-rays

The patients underwent a bi-plane X-ray (EOS Imaging Inc., Paris, France) of the lower limb with the true and misplaced external markers taped to their pelvis. The bi-plane X-ray was prescribed to the patients by their orthopaedic surgeon to plan or evaluate the surgery. The mean total dose area product was 1.9e-4 ± 1.5e-4 Gy/m$^2$, the mean total dose was 5.47e-4 ± 4.04e-4 Gy and the mean time of exposure was 16.06 ± 10.39 seconds.

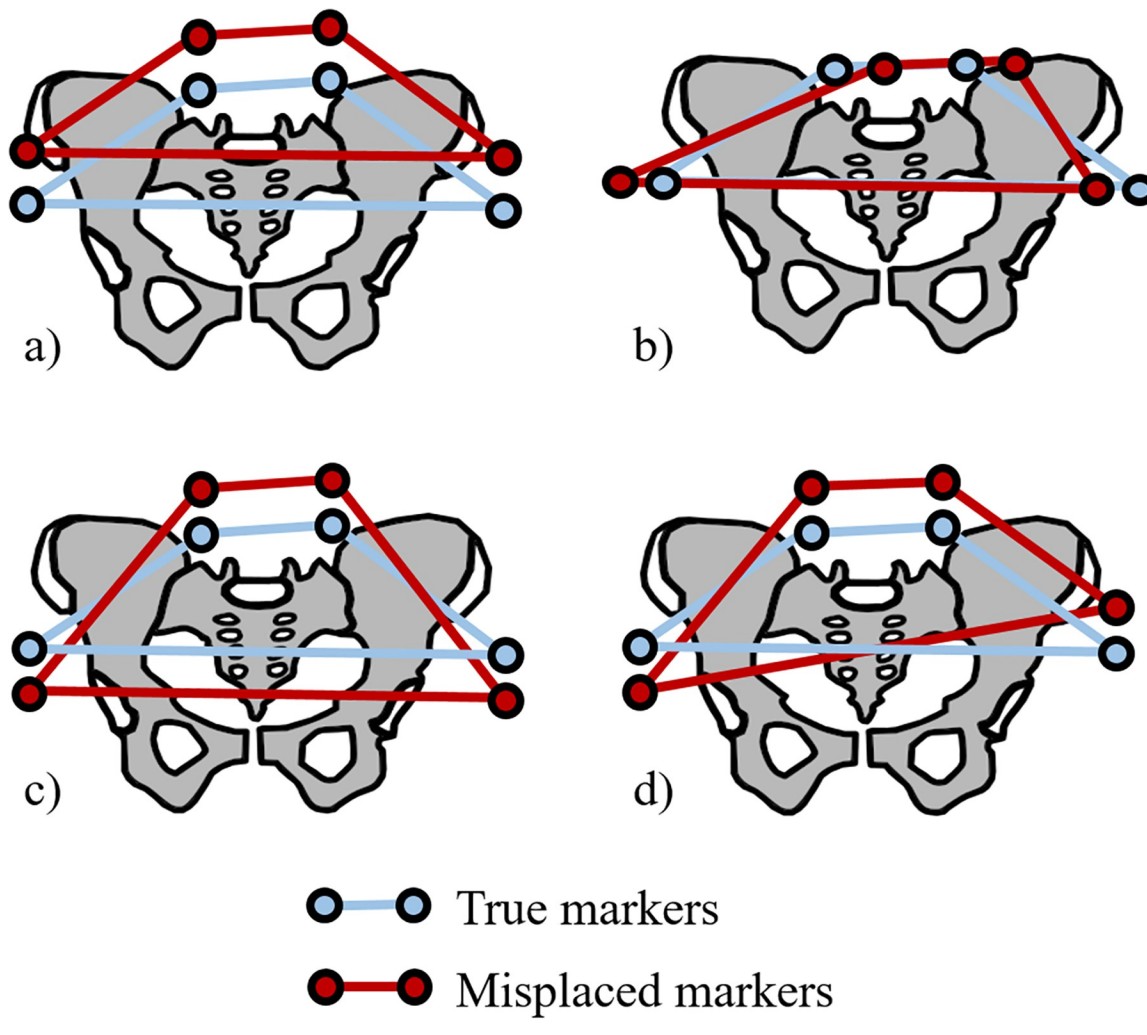

**Fig 2. External marker configurations on the pelvis: a) Vertical Offset, b) Axial Rotation, c) Anteversion, d) Obliquity.**

### Motion capture

After the bi-plane X-ray, the patients were conducted in a wheelchair to a gait laboratory with the true and misplaced external markers of the pelvis still equipped. Great care was taken to avoid any displacement of the markers during the patient's transportation. The patients were then equipped with the additional markers of the Conventional Gait Model (CGM) [22] and their gait was measured at a self-selected speed on a 10m walkway. The trajectories of the 12.5 mm skin markers were measured at 100 Hz by a 12 cameras motion capture system (Oqus7+, QTM 2.14, Qualisys, Göteborg, Sweden). The marker trajectories did not present gaps and were not filtered. Gait cycle events were identified with ground reaction forces when possible and based on the foot markers trajectories if not. The gait cycle was defined from foot strike to foot strike of the same foot. The motion capture data were processed using Matlab (R2016b, The MathWorks, Inc., Natick, Massachusetts, United States) and Biomechanical Toolkit [23] (https://gitlab.unige.ch/KLab/fusion_biplane_xrays_motion_capture).

### External and internal marker identification on bi-plane X-ray

We developed an algorithm based on the Image Processing toolbox of Matlab (R2016b, The MathWorks, Inc., Natick, Massachusetts, United States) to identify automatically the 3D position of the external markers on the DICOM images of the bi-plane X-rays (see S1 File). A second algorithm was developed to identify manually the 3D position of the internal markers on the DICOM images by fitting simple geometric element to the anatomy of the bones (see S2 File). The internal markers were: the right and left anterior superior iliac spine (RASI$_I$, LASI$_I$), the pubic symphysis (PSYM), and the right and left hip joint centre (RHJC, LHJC). Both algorithms were implemented within a Graphical User Interface (https://gitlab.unige.ch/KLab/fusion_internal_point_extraction).

### Kinematic models

The CGM was used to define the external local coordinate systems (LCS) of the pelvis (PE) and thigh (TE) [22]. The hip joint centre was either computed by regression [24] or identified on the bi-plane X-ray. The internal LCS of the pelvis (PI) was based on the plane defined by the right and left anterior superior iliac spines and the pubic symphysis (1) (see S3 File). This plane was chosen as it is used to define the cup placement in total hip arthroplasty [25]. The rotation matrix $\mathbf{R}_{X \to PI}$ from the coordinate system of the bi-plane X-ray to the internal pelvis LCS was defined as follow:

$$\mathbf{mASI}_I = (\mathbf{RASI}_I + \mathbf{LASI}_I)/2$$

$$\begin{cases} \mathbf{y}_{PI} = (\mathbf{LASI}_I - \mathbf{RASI}_I)/ \parallel (\mathbf{LASI}_I - \mathbf{RASI}_I) \parallel \\[1mm] \mathbf{z}_{tmp} = m\mathbf{ASI}_I - \mathbf{PSYM} \\[1mm] \mathbf{x}_{PI} = (\mathbf{y}_{PI} \times \mathbf{z}_{tmp})/ \parallel (\mathbf{y}_{PI} \times \mathbf{z}_{tmp}) \parallel \\[1mm] \mathbf{z}_{PI} = (\mathbf{x}_{PI} \times \mathbf{y}_{PI}) \end{cases} \tag{1}$$

$$\mathbf{R}_{X \to PI} = \begin{bmatrix} \mathbf{x}_{PI} & \mathbf{y}_{PI} & \mathbf{z}_{PI} \end{bmatrix}$$

The pelvic and hip angles were computed with a YXZ sequence of Cardan angles.

### Anatomical pelvis orientation during motion capture

After defining the orientation of the anatomical pelvis LCS ($\mathbf{R}_{X \to PI}$) and of the external marker based pelvis LCS ($\mathbf{R}_{X \to PE}$) with respect to the bi-plane X-ray coordinate system (see S3 File), the rotation matrix between the anatomical and marker based LCS ($\mathbf{R}_{PE \to PI}$) was computed as follow (2):

$$\mathbf{R}_{PE \to PI} = (\mathbf{R}_{X \to PE})^{-1} . \mathbf{R}_{X \to PI} \tag{2}$$

After the motion capture, the orientation of the marker based pelvis LCS with respect to the coordinate system of the laboratory ($\mathbf{R}_{L \to PE}$) was computed. Then, the orientation of the anatomical pelvis LCS with respect to the coordinate system of the laboratory ($\mathbf{R}_{L \to PI}$) was computed by using the orientation of the anatomical pelvis LCS with respect to the orientation of the marker based pelvis LCS as follows (3):

$$\mathbf{R}_{L \to PI} = \mathbf{R}_{L \to PE} . \mathbf{R}_{PE \to PI} \tag{3}$$

Finally, the marker based and anatomical pelvic and hip angles were computed following the CGM convention (see S3 File) (https://gitlab.unige.ch/KLab/fusion_biplane_xrays_motion_capture).

## Outcome

Two pelvis kinematics were evaluated: 1) the standard kinematics based on external markers and 2) the fusion kinematics based on internal markers. Three hip kinematics were evaluated: 1) the standard kinematics with external pelvis LCS and the hip joint centre (HJC) computed with Hara's regression equations [24], 2) the semi-fusion kinematics with the internal pelvis LCS and the HJC computed by regression and, 3) the fusion kinematics that uses the internal pelvis LCS and the HJC identified with the bi-plane X-ray. These kinematic parameters were computed for the $2^4$ possible configurations of markers (4 markers on the pelvis with 2 positions per markers: true or misplaced). The kinematics of each gait cycle was resampled to 101 points with cubic splines to express it in percentage of the gait cycle. The variability was assessed by the root mean square deviation (RMSD) and the range of variability, i.e. the range of the differences between the different configurations of markers. The intrinsic variability was removed from the analysis by computing the RMSD and range for only one gait cycle per patient. As the fusion method do not affect the extrinsic variability due to instrument errors and soft tissue artefacts, these sources of extrinsic variability are constant between the standard method and the fusion method. Consequently, the only difference in variability between the outcomes of the two methods is extrinsic and is due to the external marker misplacement error. The RMSD for the angle $\theta$ of patient $i$ ($i$ = a, b, c, d) was computed as follow (4):

$$RMSD_i^{\theta} = \sqrt{\frac{\sum_{j=1}^{n_{GC}} \left( \sum_{k=1}^{n_{MC}} \left( \sum_{t=1}^{n_f} (\theta_{jkt} - \overline{\theta_{jt}})^2 \right) \right)}{n_{GC}.n_{MC}.n_f}} \tag{4}$$

Where $n_{GC}$ is the number of gait cycles of patient $i$, $n_{MC}$ is the number of marker configurations, $n_f$ is the number of frames, $\theta_{jkt}$ is the joint angle value of gait cycle $j$ for the marker configuration $k$ at the frame $t$, and $\overline{\theta_{jt}}$ is the mean angle across marker configuration at frame $t$ for the gait cycle $j$. The range of variability for the angle $\theta$ of patient $i$ ($i$ = a,b,c,d) was computed as follow (5):

$$range_i^{\theta} = \frac{\sum_{j=1}^{n_{GC}} \left( \sum_{t=1}^{n_f} (\max(\theta_{jt}) - \min(\theta_{jt})) \right)}{n_{GC}.n_f} \tag{5}$$

Where $\theta_{jt}$ is the vector with the angle values for all marker configurations of the gait cycle $j$ at frame $t$.

The position of the HJC obtained by regression was compared to the position identified on the bi-plane X-ray. The root mean square error (RMSE) of the HJC obtained by regression was computed on each axis of the anatomical LCS (6) as well as the square root of the range of the root squared error (7).

$$RMSE_{HJC} = \sqrt{\frac{\sum_{k=1}^{n_{MC}} \left( \mathbf{HJC}_k^R - \mathbf{HJC}^A \right)^2}{n_{MC}}} \tag{6}$$

$$range_{RSE} = \sqrt{\max((\mathbf{HJC}_k^R - \mathbf{HJC}^A)^2) - \min((\mathbf{HJC}_k^R - \mathbf{HJC}^A)^2)} \tag{7}$$

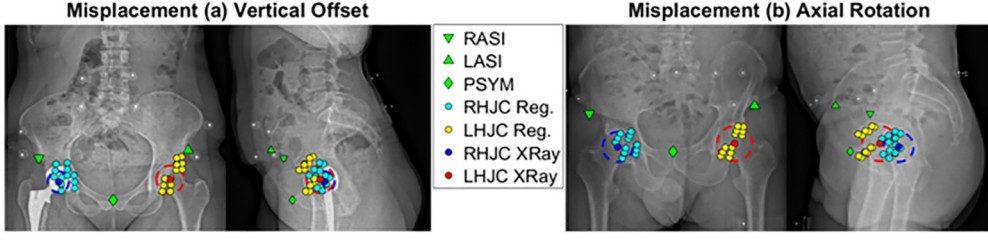

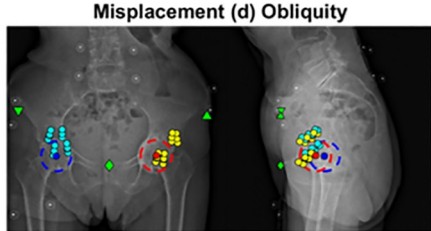

**Fig 3. Hip joint centre: Regression versus bi-plane X-ray.**

Where $n_{MC}$ is the number of marker configurations, $\mathbf{HJC}^R_k$ is the HJC position obtained with the regression for the marker configurations $k$, and $\mathbf{HJC}^A$ was the hip joint centre position identified with the bi-plane X-ray.

The effect of the fusion method was assessed with a paired Wilcoxon signed rank test for the pelvic angles and with a Kruskal-Wallis test for the hip angles. The post-hoc outcomes of the Kruskal-Wallis test were evaluated using a paired Wilcoxon signed rank test. The p-value was set to 0.05. Non-parametric statistical analysis was performed due to the low number of patients included in the study.

## Results

The mean marker misplacement error was $41 \pm 4$ mm. The mean self-selected gait speed was respectively 1.08 m.s$^{-1}$, 0.81 m.s$^{-1}$, 1.06 m.s$^{-1}$, and 0.83 m.s$^{-1}$ for patient (a), (b), (c), and (d).

### Hip position

The RMSE for marker-based HJC varied between 9.7 mm and 25.4 mm (Fig 3, Table 1). The range of the root squared error varied between 18.0 mm and 41.4 mm.

**Table 1. Hip joint centre error: Regression compared to the hip joint centre identified with bi-plane X-rays.** RMSE and range are in mm.

| | | a. Vertical offset | | b. Axial Rotation | | c. Anteversion | | d. Obliquity | |
|---|---|---|---|---|---|---|---|---|---|
| | | RMSE | Range | RMSE | Range | RMSE | Range | RMSE | Range |
| Right Hip Joint Centre | $x_{PA}$ | 14.5 | 25 | 17.8 | 29.4 | 16 | 27.5 | 11.3 | 18 |
| | $y_{PA}$ | 14.9 | 29 | 13.2 | 23.7 | 13.4 | 25 | 25.4 | 41.4 |
| | $z_{PA}$ | 9.7 | 33.8 | 12.8 | 25.6 | 10.4 | 21.8 | 21.4 | 34.6 |
| Left Hip Joint Centre | $x_{PA}$ | 11.6 | 19 | 11.6 | 21.4 | 14.3 | 23.4 | 21 | 32.6 |
| | $y_{PA}$ | 19.2 | 34.4 | 17.2 | 25.6 | 20 | 35.4 | 20 | 34.6 |
| | $z_{PA}$ | 18.3 | 31.8 | 21.5 | 32.1 | 15.8 | 29.6 | 13.9 | 25.7 |

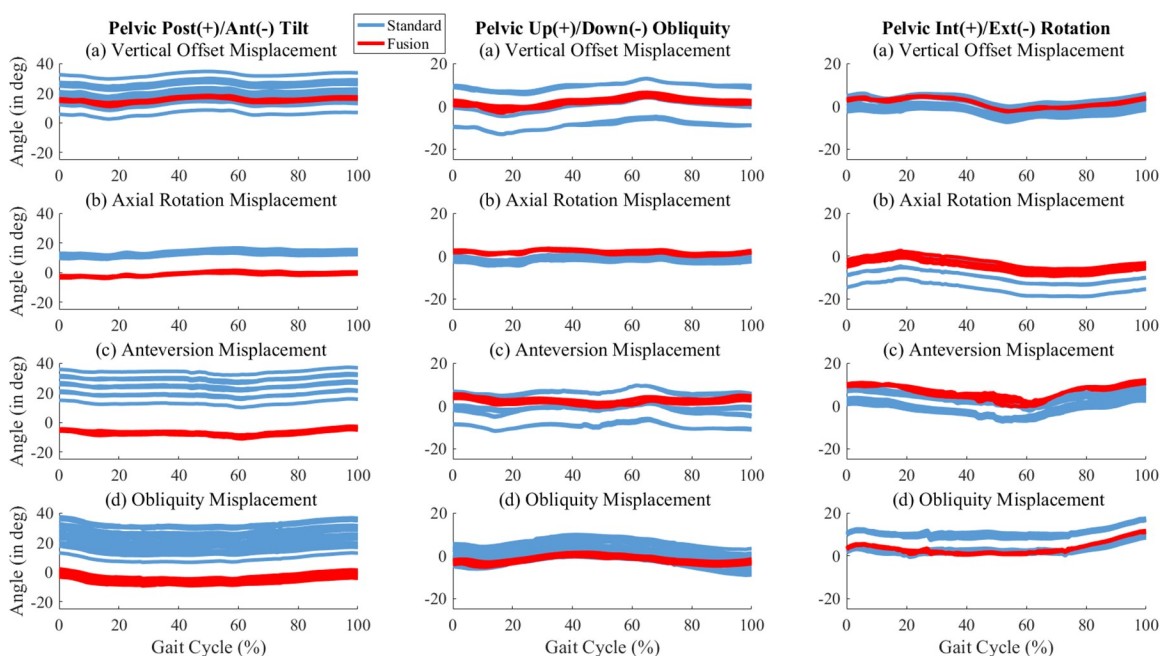

**Fig 4. Pelvis kinematics, standard method versus fusion method.** The marker misplacements are as follow: (a) Vertical Offset, (b) Axial Rotation, (c) Anterversion, (d) Obliquity.

## Pelvis kinematics

Large offsets were observed on the pelvis tilt computed from the anatomical pelvis LCS for patient (b), (c), and (d) but not for patient (a) (Fig 4). This offset was consistent with the tilt angle between the marker-based and anatomical pelvis LCS which were respectively: 5.4˚, 14.3˚, 21.2˚, and 16.0˚ for patients (a), (b), (c), and (d).

The RMSD and range of variability were significantly reduced with the fusion method ($p < 0.001$, Fig 5, Table 2). The average decrease was 3.1±1.9˚ (-78±13%) and 11.3±7.3˚ (-80±12%) for the RMSD and range of variability. The RMSD and range of variability of the axial rotation misplacement (patient (b)) were on average 3.4 times smaller than the other three misplacements.

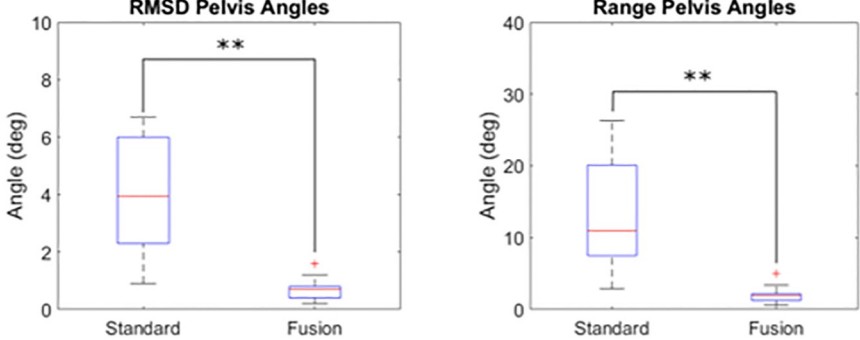

**Fig 5. Overview of the differences of variability for the pelvic angles with the standard method and fusion method** (** for $p < 0.001$).

 

**Table 2. Variability of the pelvis angles, all results are in degree.**

|  | | a. Vertical Offset | | b. Axial Rotation | | c. Anteversion | | d. Obliquity | |
|---|---|---|---|---|---|---|---|---|---|
|  | Method | RMSD | Range | RMSD | Range | RMSD | Range | RMSD | Range |
| **Tilt** | Standard | 6.7 | 26.3 | 0.9 | 3.3 | 5.4 | 21.4 | 6.4 | 24 |
|  | Fusion | 0.5 | 1.7 | 0.4 | 1.4 | 0.6 | 2.2 | 1.6 | 5 |
| **Obliquity** | Standard | 6.5 | 18.8 | 1 | 2.9 | 5.6 | 16.2 | 3.2 | 10.5 |
|  | Fusion | 0.8 | 2 | 0.4 | 1.1 | 0.8 | 2.2 | 0.8 | 2 |
| **Rotation** | Standard | 2 | 7.3 | 4.1 | 11.5 | 2.6 | 7.7 | 3.8 | 9.8 |
|  | Fusion | 0.2 | 0.6 | 1.2 | 3.4 | 0.8 | 1.9 | 0.3 | 0.8 |

## Hip kinematics

The hip flexion/extension angle computed with the internal pelvis LCS showed large offsets for patient (b), (c), and (d) which led to average peak extension of -9.3±0.6˚, -26.3±0.9˚, and -19.0 ±1.7˚ respectively (Fig 6). Patient (a) did present a smaller offset (peak extension of 3.1±0.6˚).

The RMSD and range of variability were significantly reduced for the fusion and semi-fusion methods (p < 0.001, Fig 7, Table 3). The semi-fusion method decreased the RMSD and range of variability by 3.3±2.2˚ (-55±36%) and 11.0±8.5˚ (-55±34%) in average. The average decrease for the fusion method was 4.1±1.9˚ (-78±18%) and 13.7±7.6˚ (-78±20%) for the RMSD and range of variability respectively. The RMSD and range of variability of the fusion method were statistically lower (p = 0.006, p = 0.002) than the semi-fusion method. The average decrease was 0.8±0.8˚ (-36±45%) for the RMSD and 2.7±2.5˚ (41±40%) for the range of variability.

On average, the RMSD was reduced by -78±15% and the range of variability was reduced by -80±16% for the pelvis and hip kinematics.

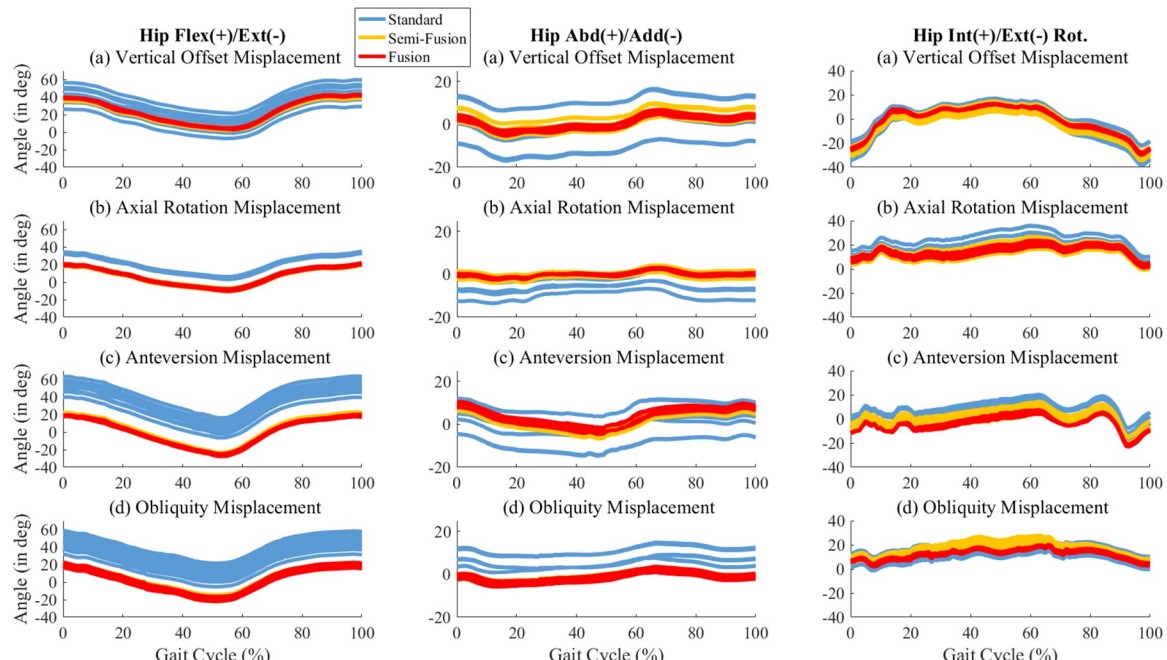

**Fig 6. Right hip kinematics with the standard method, semi-fusion method and fusion method.** The marker misplacements is as follow: (a) Vertical Offset, (b) Axial Rotation, (c) Anterversion, (d) Obliquity.

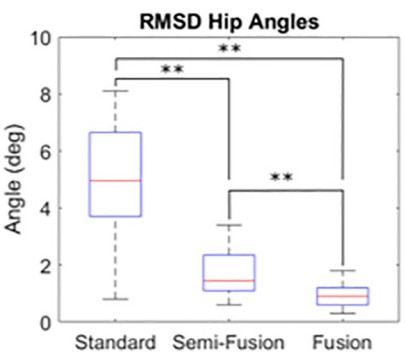 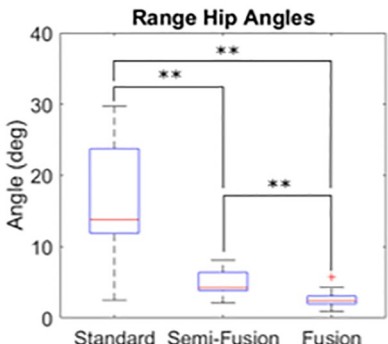

**Fig 7. Overview of the differences of variability for the hip angles with the standard method, the semi-fusion method, and the fusion method.** (** for p < 0.01).

**Table 3. Variability of the hip joint angles, all results are in degree.**

|  | Method | a. Vertical Offset | | b. Axial Rotation | | c. Anteversion | | d. Obliquity | |
|---|---|---|---|---|---|---|---|---|---|
|  |  | RMSD | Range | RMSD | Range | RMSD | Range | RMSD | Range |
| **Flexion—Extension** | *Standard* | 7.5 | 29.7 | 0.8 | 2.6 | 6 | 23.9 | 7.1 | 26.6 |
|  | *Semi-Fusion* | 1.1 | 4 | 1.1 | 3.3 | 1.3 | 4.6 | 1.1 | 4 |
|  | *Fusion* | 0.6 | 1.8 | 0.6 | 2.1 | 0.8 | 2.6 | 1.8 | 5.8 |
| **Adduction—Abduction** | *Standard* | 8 | 23 | 3 | 8.3 | 6.2 | 17.8 | 4.6 | 13.2 |
|  | *Semi-Fusion* | 2.1 | 6.5 | 1.2 | 3.3 | 1.4 | 4 | 0.7 | 2 |
|  | *Fusion* | 0.9 | 2.2 | 0.3 | 0.9 | 1.1 | 2.8 | 1 | 2.6 |
| **Internal—External Rotation** | *Standard* | 3.9 | 11.9 | 5.5 | 14.8 | 4 | 11.5 | 3.6 | 12.3 |
|  | *Semi-Fusion* | 2.5 | 8 | 3.5 | 8.5 | 2.7 | 8.5 | 1.9 | 6.8 |
|  | *Fusion* | 0.5 | 1.5 | 1.6 | 4.4 | 0.9 | 2.4 | 0.8 | 2.4 |

## Discussion

The goal of this study was to evaluate whether the fusion of motion capture and 3D medical imaging could reduce the extrinsic variability of the pelvis and hip kinematics due to external marker misplacements in clinical gait analysis. A controlled marker misplacement error was introduced in multiple measurements of the pelvis and hip kinematics during gait to assess the reduction of variability accountable to the fusion method.

The fusion method significantly reduced the extrinsic variability of gait kinematics due to marker misplacements but not entirely. A perfect correction would have led to a RMSD and range of 0 degree per gait cycle. The residual variability was likely due to the differences in soft tissue artefacts between the true external markers and the misplaced external markers. Indeed, variations in marker placement lead to variations in soft tissue artefacts that lead to variations in kinematic patterns. As the fusion method only applies a constant offset to the angles computed from the skin markers, the variations of kinematic pattern due to soft tissue artefacts are not corrected. A solution to this issue would be to identify a technical markerset with minimal soft tissue artefacts and combine it with the present method.

Interestingly, the variability induced by the axial rotation misplacement (patient (b)) was much lower than the other cases of misplacements. This suggests that marker misplacements errors are not isotropic, i.e. some directions of misplacement lead to larger errors than others. An extended knowledge on this phenomenon should prove useful when defining palpation methods for skin marker placement.

Various methods were previously developed to tackle the variability due to skin marker misplacements. Gorton et al. [7] showed that a standardised marker placement method could reduce the inter-assessor variability. However, while their method decreased the variability of multiple kinematic parameters, it also increased the variability between assessors for the pelvic tilt (+4% to +8%) and the hip rotation (+19%). Marker placement devices could reduce the greatest absolute difference between days by 8 degrees for the hip flexion angle [12] and the RMS within and between operator by 46% and 37% respectively [13]. The drawbacks of these devices are that 1) the patient should "stand still for up to 10 min" [12] which seems impractical in clinical settings and 2) this device cannot be used for patients that had changes in their anatomy, e.g. patients that underwent orthopaedic surgery or children with cerebral palsy that grew up. The real-time feedback system developed by Macaulay et al. [11] could reduce the variability (95% confidence interval) between novice examiners by 47% for the hip frontal plane and 56% for the hip flexion angle in an healthy subjects. Although promising, this device has not yet been tested for a pathological population.

In our study, the extrinsic variability was more minimised than in the above mentioned methods. Indeed, the RMSD was on average reduced by -78±15% and the range of variability was reduced by -80±16% for the pelvis and hip kinematics. Still, comparison with the previously reported methods is limited since we did not perform a test re-test analysis to avoid unnecessary radiation exposure for the patients.

The marker misplacement magnitude in our study (41mm) was slightly higher than the maximal intra-examiner and inter-examiner precision reported by Della Croce et al. [5] (21mm and 24.7mm respectively for the placement of pelvic markers). We chose a space of two skin markers between the true external marker and the misplaced external marker to avoid measurement errors during motion capture and as a worst-case scenario to show the robustness of the method.

The bi-plane X-ray method suffers from several limitations in the context of the fusion with motion capture. First, this device is costly and will not be available for all gait laboratories. Second, the low-dose radiations, although beneficial for the patients' health, can lead to a low contrast on the X-rays images. Consequently, anatomical points can be difficult to identify, especially in an elderly population. This process introduces a new source of extrinsic variability that should be assessed in further studies. Third, it can be challenging for the patients to stand perfectly still during an acquisition that lasts between 10 to 15 seconds. Motion of the patient during the acquisition leads to inaccuracies on the X-ray images. Fourth, the field of acquisition is limited to a width of 0.34m and 0.32m for the frontal and sagittal view at the centre of the cabin. Thus, skin markers taped on the lateral side of the patients are often not visible. An alternative would be to use a different markerset or a different imaging technique such as 3D ultrasound [14].

Regarding the kinematic outcome of the fusion method, the hip kinematics presented high peak hip extension angles compared to the standard method. These peaks were due to the tilt offset between the external pelvis LCS and the internal pelvis LCS. The tilt offset could be affected by two sources: 1) the pelvic anatomy or 2) inaccuracies during the identification of the anterior superior iliac spines on the X-ray images. Indeed, these internal markers can be difficult to identify accurately on the sagittal view image. The measurement of the distance between those points with bi-plane X-rays was previously validated with a dry pelvic bone [26]. However, the contrast on the image was higher than what we observed for our patients. Thus, the accuracy and repeatability of our anatomical point identification algorithm still needs to be assessed.

The next question that arises from the high hip extension angles is whether the anterior pelvic plane is a relevant reference for the hip and pelvis kinematics or if another plane should be

used. Indeed, a similar outcome on variability should occur with a different set of internal markers. The main criterion is that the internal markers should be clearly visible on both the frontal and sagittal views and make sense from an anatomical point of view. As an example, the posterior superior iliac spines classically used for external markers were hardly visible on the sagittal X-ray and could not be used.

Finally, this study had a low number of participants but we believe that the sample is sufficient to prove the concept of the method and its potential.

## Conclusion

The fusion of motion capture and 3D medical imaging significantly reduced the extrinsic variability due to skin marker misplacement for the pelvis and hip kinematics during gait. These results are promising and the fusion method could be extended to other joints and movements. However, the identification of the anatomical landmarks on the bi-plane X-rays represent a new source of extrinsic variability that should be assessed before applying this method in clinical settings.

## Supporting information

**S1 File. Detection of external markers.**
(PDF)

**S2 File. Identification of internal markers.**
(PDF)

**S3 File. Kinematics computation.**
(PDF)

**S4 File. Pelvic kinematics for each marker configuration of patient (a).**
(PDF)

## Acknowledgments

The authors wish to thank Florent Moissenet and Sandra Komarzynski for their help in the development of the internal marker identification tool.

## Author Contributions

**Conceptualization:** Xavier Gasparutto, Stéphane Armand.

**Data curation:** Jennifer Wegrzyk, Kevin Rose-Dulcina.

**Formal analysis:** Xavier Gasparutto, Stéphane Armand.

**Funding acquisition:** Didier Hannouche.

**Investigation:** Xavier Gasparutto.

**Methodology:** Xavier Gasparutto.

**Software:** Xavier Gasparutto.

**Supervision:** Stéphane Armand.

**Writing – original draft:** Xavier Gasparutto.

**Writing – review & editing:** Jennifer Wegrzyk, Kevin Rose-Dulcina, Didier Hannouche, Stéphane Armand.

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
