## [Decision Letter · Decision Letter 0]

9 Oct 2019

PONE-D-19-19297

Can the fusion of motion capture and 3D medical imaging reduce the extrinsic variability due to marker misplacements?

PLOS ONE

Dear Dr Gasparutto,

Thank you for submitting your manuscript to PLOS ONE. After careful consideration, we feel that it has merit but does not fully meet PLOS ONE’s publication criteria as it currently stands. Therefore, we invite you to submit a revised version of the manuscript that addresses the points raised during the review process.

I read the reviewers' comments, which were highly constructive.  I also read the manuscript myself, as the topic is within my own expertise.  I recommend you take a look at two methodological papers on evaluating surface-based motion capture against dynamic X-ray imaging (Li et al., 2012, Journal of Biomechanics) and integrating the two approaches (Zheng et al., 2014, Journal of Biomechanics).

We would appreciate receiving your revised manuscript by Nov 23 2019 11:59PM. To enhance the reproducibility of your results, we recommend that if applicable you deposit your laboratory protocols in protocols.io, where a protocol can be assigned its own identifier (DOI) such that it can be cited independently in the future. For instructions see: http://journals.plos.org/plosone/s/submission-guidelines#loc-laboratory-protocols

We look forward to receiving your revised manuscript.

Kind regards,

Xudong Zhang

Academic Editor

PLOS ONE

**Journal Requirements:**

**Comments to the Author**

1. Is the manuscript technically sound, and do the data support the conclusions?

Reviewer #1: Yes

Reviewer #2: Yes

2. Has the statistical analysis been performed appropriately and rigorously? 

Reviewer #1: Yes

Reviewer #2: N/A

3. Have the authors made all data underlying the findings in their manuscript fully available?

Reviewer #1: No

Reviewer #2: Yes

4. Is the manuscript presented in an intelligible fashion and written in standard English?

Reviewer #1: Yes

Reviewer #2: Yes

5. Review Comments to the Author

Reviewer #1: In the submitted manuscript, the authors present an investigation into the feasibility of fusing optical motion capture with bi-plane X-rays to reduce kinematic variability due to misplaced skin markers (vertical offset, axial rotation, anteversion, and obliquity). In this study, they specifically focused on pelvis and hip kinematics during a preferred-speed walking task completed by four subjects. Hip joint centers (HJCs) were estimated by either regression with the optical motion capture data or manually by the bi-plane X-rays. Pelvis kinematics were estimated using a pelvic anatomical frame defined by either the external markers for optical motion capture (standard) or the internal markers provided by the bi-plane X-rays (fusion). Hip kinematics were estimated using standard pelvis kinematics + regression-based HJC, fusion pelvis kinematics + regression-based HJC, and fusion pelvis kinematics + X-ray-based HJC. Nonparametric statistical analyses were conducted for comparing the different pelvis and hip kinematics. The supporting information files were excellent and clear.

The main weakness of this paper is missing or unclear information. For example, it was not clear until the results that each patient effectively represents a different kind of marker misplacement, meaning the markers for patient (a) were given a vertical offset, the markers for patient (b) were given an axial rotation, and so on. Next, the authors offer no explanation for why they are conducting nonparametric statistical analyses (I assume it is because the data are non-normally distributed, in which case they should state as much). Also, the figure captions for Figs. 3, 4, and 6 need to be more descriptive. For example, in Fig. 3, what do the dashed lines around the hip joint centers identified by the X-ray mean, why are there 12 results for the HJC regression for each hip, specify again that each subfigure is a different patient, etc. Also, in reference to Figs. 4 and 6, could the authors provide some insight in the main body of the paper as to why some subplots appear to have discrete bands of results (see, for example, the plot for Pelvic Post(+)/Ant(-) Tilt for Anteversion Misplacement)? Finally, the sentence on Lines 189-190 feels likely out of place. It would be better situated in the “Bi-plane X-Rays” section starting on Line 98.

This study was well-designed and the paper was fairly well-written. As such, it is the opinion of this reviewer that it should be accepted with minor revisions. However, the authors are encouraged to carefully proofread the paper when addressing the previous comments (for example, I believe R_L->PM on Line 141 should be R_L->PE and “wish” on line 322 is missing the “w”).

Reviewer #2: Reducing the extrinsic vrability occurred from marker misplacement is an essential issue for current motion studies, which is still lay hopes on an effective method. Many researches attempted analyzing the motion data with assisted medical imaging technology, but the effectiveness is weak. This work made a progress of solving this problem by using a fusion way of capturing motion and bi-plane X-ray imaging. This could capture the bone motion rather than the traditional sufrace motion. The work is technically sound. However, using bi-plane X-ray imaging is not that innovative, and no modelling was introduced in the work. The iilustration are unclear with blurred figures. The literature reviews and their references are not up to date.

6. PLOS authors have the option to publish the peer review history of their article (what does this mean?). If published, this will include your full peer review and any attached files.

Reviewer #1: No

Reviewer #2: No

---

## [Author Response · Author response to Decision Letter 0]

24 Oct 2019

Dear Dr Zhang,

We would like to thank you and the reviewers for your time and for the constructive feedback that helped improve the quality of the paper. Thank you also for sharing references with us, we added Zheng et al. (2014) as a reference to complete the introduction of the paper. We will share our data on Yareta and our codes on GitLab. Only one part of the code is currently shared, we are currently preparing the rest of the code and data used in the paper.

Please find the answer to the reviewers in red in the text below.

Review Comments to the Author

Reviewer #1: In the submitted manuscript, the authors present an investigation into the feasibility of fusing optical motion capture with bi-plane X-rays to reduce kinematic variability due to misplaced skin markers (vertical offset, axial rotation, anteversion, and obliquity). In this study, they specifically focused on pelvis and hip kinematics during a preferred-speed walking task completed by four subjects. Hip joint centres (HJCs) were estimated by either regression with the optical motion capture data or manually by the bi-plane X-rays. Pelvis kinematics were estimated using a pelvic anatomical frame defined by either the external markers for optical motion capture (standard) or the internal markers provided by the bi-plane X-rays (fusion). Hip kinematics were estimated using standard pelvis kinematics + regression-based HJC, fusion pelvis kinematics + regression-based HJC, and fusion pelvis kinematics + X-ray-based HJC. Nonparametric statistical analyses were conducted for comparing the different pelvis and hip kinematics. The supporting information files were excellent and clear.

Authors answer: “Thank you for those comments.”

The main weakness of this paper is missing or unclear information. For example, it was not clear until the results that each patient effectively represents a different kind of marker misplacement, meaning the markers for patient (a) were given a vertical offset, the markers for patient (b) were given an axial rotation, and so on. 

Authors answer: “We added a sentence to the Material & Method section to clarify this point:

L.91 “Each patient had one configuration of marker misplacement””

Next, the authors offer no explanation for why they are conducting nonparametric statistical analyses (I assume it is because the data are non-normally distributed, in which case they should state as much). 

Authors answer: “The number of patients was too low to use parametric statistical analysis. The following sentence was added to the method section:

L.187: “Non-parametric statistical analysis was performed due to the low number of patients included in the study.””

Also, the figure captions for Figs. 3, 4, and 6 need to be more descriptive. For example, in Fig. 3, what do the dashed lines around the hip joint centres identified by the X-ray mean, why are there 12 results for the HJC regression for each hip, specify again that each subfigure is a different patient, etc. 

Authors answer: “A sphere fitting of the femoral head was performed on the bi-plane x-ray to identify the hip joint centres. The dashed lines represent this sphere projected on both planes, i.e. the dimension of the femoral head. For each HJC obtained with regression, there are actually 16 points (one for each combination of markers) but the points overlap.

We added the following sentence to the description of Figure 3:

“Each figure represent one patient with a different marker misplacement. The points represent the hip joint center location and the dashed circles represent the size of the femoral head identified by sphere fitting.””

Also, in reference to Figs. 4 and 6, could the authors provide some insight in the main body of the paper as to why some subplots appear to have discrete bands of results (see, for example, the plot for Pelvic Post(+)/Ant(-) Tilt for Anteversion Misplacement)? 

Authors answer: “All subplots have discrete bands of results but some are more striking than others (as for your example). The misplacement error introduced in the study was discrete and rather large (4cm) which leads to this discrete pattern. The way curves are grouped depend on the marker configurations. As an example, for the pelvic obliquity and pelvic rotation, each group is associated with a combination of the ASI markers (either 2 palpated, 2 misplaced or one of each). 

For your example (Pelvic Post(+)/Ant(-) Tilt for Anteversion Misplacement) there is 5 different groups also associated to a specific combination of markers that we describe below (from highest tilt to lowest tilt)

- Group 1: all markers are the misplaced ones (1 curve) – maximal tilt

- Group 2: all markers but one are misplaced (4 curves)

- Group 3: 2 markers misplaced and 2 palpated markers (6 curves)

- Group 4: all markers but one are palpated (4 curves)

- Group 5: all markers are palpated (1 curve) – minimal tilt

However, the groups seen on the tilt for the other misplacement are different, those groups depend on the misplacements. To summarize, some combinations of skin markers maximize the values on one angle, others minimize it. In between, there are multiple combinations that lead to similar values of joint angles. As the misplacement was discrete, the results shows discrete curves. We added a figure in Supplementary Material to clarify this issue. It depicts every marker combination with a different colour and the associated pelvic angles are plotted with the same colour and make it easier to understand what we did. 

If you find it appropriate it could be added to the main body of the paper but it might be too much figures.”

Finally, the sentence on Lines 189-190 feels likely out of place. It would be better situated in the “Bi-plane X-Rays” section starting on Line 98.

Authors answer: “This sentence was moved to the paragraph “Bi-plane X-rays” in the method section.”

This study was well-designed and the paper was fairly well-written. As such, it is the opinion of this reviewer that it should be accepted with minor revisions. However, the authors are encouraged to carefully proofread the paper when addressing the previous comments (for example, I believe R_L->PM on Line 141 should be R_L->PE and “wish” on line 322 is missing the “w”).

Authors answer: “R_L->PM was changed to R_L->PE and the missing “w” was corrected.”

Reviewer #2: Reducing the extrinsic vrability occurred from marker misplacement is an essential issue for current motion studies, which is still lay hopes on an effective method. Many researches attempted analyzing the motion data with assisted medical imaging technology, but the effectiveness is weak. This work made a progress of solving this problem by using a fusion way of capturing motion and bi-plane X-ray imaging. This could capture the bone motion rather than the traditional sufrace motion. The work is technically sound. However, using bi-plane X-ray imaging is not that innovative, and no modelling was introduced in the work. 

Authors answer: “Thank you for those comments. We do not claim to introduce a novel method. As underlined in the introduction (L70-72), the point of the paper is to demonstrate that the fusion of motion capture and medical imaging could help reducing the variability due to marker misplacement. This could work with a different medical imaging method, as suggested in the discussion (L299 - 300). No specific modelling was required for this study.”

The illustration are unclear with blurred figures. 

Authors answer: “This is due to the pdf generation of PlosONE, there is a link on the top right of the pdf to download the figures with a higher quality.”

The literature reviews and their references are not up to date.

Authors answer: “We added the following references to the papers’ introduction:

- Akpinar B, Thorhauer E, Tashman S, et al. Tibiofemoral Cartilage Contact Differences Between Level Walking and Downhill Running. Orthop J Sport Med 2019; 7: 1–7.

- D’Isidoro F. How does the hip joint move? Techniques and applications. ETH Zurich, 2018.

- Zheng L, Li K, Shetye S, et al. Integrating dynamic stereo-radiography and surface-based motion data for subject-specific musculoskeletal dynamic modeling. J Biomech 2014; 47: 3217–3221.

- Valente G, Crimi G, Vanella N, et al. nmsBuilder: Freeware to create subject-specific musculoskeletal models for OpenSim. Comput Methods Programs Biomed 2017; 152: 85—92.”

---

## [Decision Letter · Decision Letter 1]

4 Dec 2019

Can the fusion of motion capture and 3D medical imaging reduce the extrinsic variability due to marker misplacements?

PONE-D-19-19297R1

Dear Dr. Gasparutto,

We are pleased to inform you that your manuscript has been judged scientifically suitable for publication and will be formally accepted for publication once it complies with all outstanding technical requirements.

With kind regards,

Xudong Zhang

Academic Editor

PLOS ONE

Additional Editor Comments (optional):

Reviewers' comments:

Reviewer's Responses to Questions

**Comments to the Author**

1. If the authors have adequately addressed your comments raised in a previous round of review and you feel that this manuscript is now acceptable for publication, you may indicate that here to bypass the “Comments to the Author” section, enter your conflict of interest statement in the “Confidential to Editor” section, and submit your "Accept" recommendation.

Reviewer #1: All comments have been addressed

Reviewer #2: All comments have been addressed

2. Is the manuscript technically sound, and do the data support the conclusions?

Reviewer #1: Yes

Reviewer #2: Yes

3. Has the statistical analysis been performed appropriately and rigorously? 

Reviewer #1: Yes

Reviewer #2: Yes

4. Have the authors made all data underlying the findings in their manuscript fully available?

Reviewer #1: Yes

Reviewer #2: Yes

5. Is the manuscript presented in an intelligible fashion and written in standard English?

Reviewer #1: Yes

Reviewer #2: Yes

6. Review Comments to the Author

Reviewer #1: The authors did a good job to revise the manuscript. In particular, the addition of supplemental document #4 is appreciated and will offer valuable insight for future readers.

Reviewer #2: I understand you do not claim to introduce a novel method because the paper focus on demonstrating the fusion of motion capture and medical imaging that could help reducing the variability due to marker misplacement. A specific modelling is still strong recommend for this study to support the investigation.

The illustration with a higher quality were found.

The literature reviews and their references are dated.

7. PLOS authors have the option to publish the peer review history of their article (what does this mean?). If published, this will include your full peer review and any attached files.

Reviewer #1: No

Reviewer #2: No

---

## [Editor Report · Acceptance letter]

14 Jan 2020

PONE-D-19-19297R1 

Can the fusion of motion capture and 3D medical imaging reduce the extrinsic variability due to marker misplacements? 

Dear Dr. Gasparutto:

I am pleased to inform you that your manuscript has been deemed suitable for publication in PLOS ONE. Congratulations! Your manuscript is now with our production department. 

With kind regards,

on behalf of

Dr. Xudong Zhang 

Academic Editor

PLOS ONE